# Beyond Semantics: Rediscovering Spatial Awareness in Vision-Language Models

## Abstract

Vision–Language Models (VLMs) excel at identifying and describing objects but often fail at spatial reasoning. We study why VLMs, such as LLaVA, under-utilize spatial cues despite having positional encodings and spatially rich vision-encoder features. Our analysis reveals a key imbalance: vision token embeddings have much larger norms than text tokens, suppressing LLM's position embedding. To expose this mechanism, we developed three interpretability tools: (1) the Position Sensitivity Index, which quantifies reliance on token order, (2) the Cross-Modality Balance, which reveals attention head allocation patterns, and (3) a RoPE Sensitivity probe, which measures dependence on rotary positional embeddings. These tools uncover that vision tokens and system prompts dominate attention. We validated our mechanistic understanding through targeted interventions that predictably restore positional sensitivity. These findings reveal previously unknown failure modes in multimodal attention and demonstrate how interpretability analysis can guide principled improvements. We will release code upon publication.

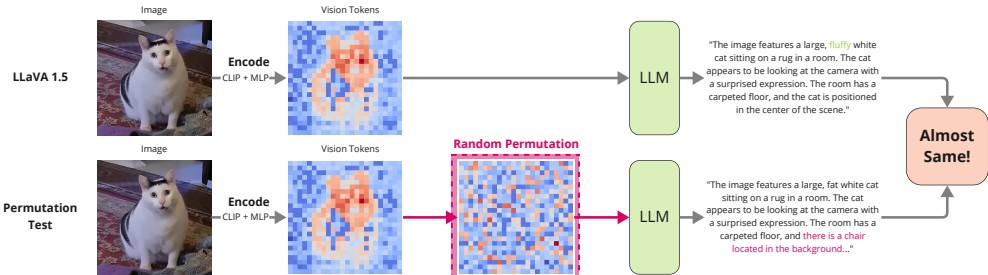

Figure 1: **Permutation Test**: Original (top) vs. randomly permuted vision tokens (bottom). Despite losing spatial ordering, the LLM accurately responds to the prompt "Describe the image," demonstrating strong robustness and a notable "bag-of-tokens" tendency, losing spatial relationships. Token embeddings are visualized using cosine similarity relative to a reference token.

## 1 Introduction

Humans process vision through two distinct pathways: the ventral stream, which identifies objects, and the dorsal stream, which encodes spatial relationships. The dorsal stream, located in the parietal lobe, is essential for processing spatial relationships, allowing us to understand where objects are and how they relate to each other. Not surprisingly, patients with damage to the parietal lobe exhibit asymmetric visual processing: they retain the ability to recognize and describe objects but lose spatial reasoning capabilities (Goodale & Milner, 1992).

Recent observations suggest that generative Vision-Language Models (VLMs) exhibit a similar asymmetry. Despite remarkable performance in object-centric (ventral) tasks such as object recognition and image captioning (Goyal et al., 2017; Antol et al., 2015; Gurari et al., 2018), VLMs struggle with even fundamental spatial reasoning queries (e.g., distinguishing "left" from "right") (Kamath et al., 2023; Tong et al., 2024b). This leads to a critical interpretability question: *Where does "space" live in VLMs, and why is it under-used?*

In this paper, we only focus on the common architecture that couples a pre-trained vision encoder with a pre-trained language model (LLM). In principle, both sides provide spatial signals: vision encoders carry layout through their feature maps and token order, while the LLM's positional mechanism (RoPE) can make the decoder sensitive to the token order. Yet the combined system often behaves like a "bag of words", mostly losing spatial relationships (Yuksekgonul et al., 2022). So the question is: **Why does the integration fail?**

We hypothesize two causes: First, the RoPE is under-utilized. Second, the last few layers of the vision encoder might not contain enough spatial information. We first diagnose and show empirical evidence of the norm mismatch of the two modalities (vision and text) causing attention imbalance. Then we develop a compact interpretability toolkit and two mechanism-aligned interventions. Our toolkit comprises: (i) a Position Sensitivity Index (PSI) tool that quantifies order use, (ii) a Cross-Modality Balance (CMB) tool to analyze head-level vision vs. text attention with prompt isolation, and (iii) a RoPE sensitivity probe tool that measures intrinsic response to positional phase shifts. We also introduce a controlled synthetic benchmark (2DS) to better measure these three indicators.

We then apply two minimal interventions to test our hypothesis: *Normalize* RMS scale-matching of vision embeddings post-projector, and *Normalize+Multilayer* combination of intermediate encoder layers. We do an extensive comparison with our interpretability tools to understand how our interventions can make the model behave differently. Our aim is understanding and measurement, instead of advancing SOTA.

**Contributions.** (1) *Diagnosis:* we identify and validate a concrete mechanism —embedding-norm skew— that links modality scale to diminished positional sensitivity in the LLM. (2) *Toolkit & data:* PSI, CMB, and RoPE probe, together with a spatially focused dataset *2DS* offer complementary views of spatial use. (3) *Mechanism-aligned probes:* we demonstrate that norm-induced problem is the causal mechanism behind positional insensitivity and that intermediate layers of the vision encoder can contribute more to spatial understanding.

## 2 RELATED WORK

**Vision-Language Models** Recent Vision-Language Models (VLMs) combine pretrained vision encoders and large language models (LLMs) to perform image understanding tasks, such as image captioning and visual question answering (Liu et al., 2023; 2024; Li et al., 2023a; Alayrac et al., 2022; Zhu et al., 2024; Shao et al., 2023; Ding et al., 2022; Hu et al., 2022). Many approaches leverage pretrained visual encoders, notably CLIP (Radford et al., 2021), and subsequently introduce an adapter module to integrate vision embeddings into pretrained LLMs, such as Vicuna (Chiang et al., 2023), enabling coherent textual outputs (Liu et al., 2023; Zhu et al., 2024). While these methods achieve impressive performance in descriptive tasks, they often struggle with spatial reasoning (Tong et al., 2024b; Yuksekgonul et al., 2023). To address spatial limitations, recent efforts have focused on augmenting training datasets with extensive spatial annotations (Tong et al., 2024a; Chen et al., 2024a; Cheng et al., 2024; Trabelsi et al., 2025). In contrast, our approach does not rely on additional spatial labels or modifications to the original training data; instead, we introduce representation-level interventions, including embedding norm normalization and mid-layer feature extraction, which demonstrate significant improvements in VLMs' spatial reasoning capabilities.

**VLM Interpretability** Interpretability of Vision-Language Models has attracted increasing attention, with the aim of explaining the internal mechanisms driving their successes or failures. Neo et al. (2025) show that vision tokens in LLaVA (Liu et al., 2023) have strong semantic properties. Concurrently, multiple studies have highlighted the consistent failures of VLMs to comprehend simple spatial relations such as "above," "below," "left," and "right" (Yuksekgonul et al., 2023; Kamath et al., 2023; Tong et al., 2024b; Chen et al., 2025). These findings underscore a critical need for enhanced interpretability and robust spatial grounding in VLM architectures. Our work investigates the underlying reasons why spatial information is overshadowed in LLMs, offering minimal yet effective interventions that notably strengthen the spatial reasoning ability.

## 3 MOTIVATION

Table 1: Impact of random vision token permutation on spatial reasoning accuracy (%).

| Dataset | Original | Permutation | Difference |
|---|---|---|---|
| VQAv2 | 78.2 | 77.35 | -0.85 |
| POPE | 87.3 | 87.10 | -0.2 |
| GQA | 61.36 | 58.62 | -2.74 |
| CV-Bench 2D | 56.59 | 56.26 | -0.33 |

VLMs demonstrate excellent capabilities in recognizing objects within images (ventral tasks) yet consistently struggle with spatial reasoning (dorsal tasks). Open-source VLMs such as LLaVA (Liu et al., 2023) typically consist of a vision encoder and an LLM. One central question we want to know is: *How do VLMs use the spatial information?* Conceptually, there are two sources for the spatial information: **vision encoder** and **LLM position embedding**. The vision encoder encodes spatial information within vision tokens. LLM position embedding directly represents the spatial information of the input tokens' order. We are particularly interested in how much the LLM leverages position embedding for spatial information.

To understand the positional sensitivity, we use LLaVA 1.5 7B (Liu et al., 2023; 2024) to randomly permute vision token embeddings after processing by the vision encoder and the MLP projection layers, but right before they are fed into the LLM (Figure 1). It effectively changes the position embedding for each token. The performance is then compared against the original unpermuted baseline. We evaluate performance on datasets including VQAv2 (Goyal et al., 2017), POPE (Li et al., 2023c), GQA (Hudson & Manning, 2019), and CV-Bench (Tong et al., 2024a). As shown in Table 1, random token permutation leads only to minor performance drops (ranging from 0.2% to 2.74%). This surprisingly small decline reveals that current VLMs evaluated with the corresponding benchmarks are largely insensitive to positional information.

Another way to investigate the role of position is to eliminate the position embedding completely. Recent work showed VLMs focus on object semantic tokens (Neo et al., 2025), and we want to know how much spatial information is encoded into the vision tokens and how much is the result of the LLM position embedding. We progressively compress the vision tokens, effectively removing the position embedding's effect on the spatial information, including detailed positional and geometric relationships among image patches. We aim to push it to the extreme to see how much information is relied on the LLM position embedding or if they are already encoded in the vision token. Using the LLaVA 1.5 7B model, we retrained with vision token counts of 256, 64, 16, and 1 (originally 576), using average pooling after the MLP projection layer. As illustrated in Figure 2, reducing the vision tokens from 576 down to just 1 for standard benchmarks results in a minor accuracy drop (-8.5% the worst case). Despite losing approximately 99.8% (from 576 to 1) of spatial resolution, the model's performance remains robust. Concurrent work also reached similar conclusion and showed robustness of compression (Chai et al., 2025).

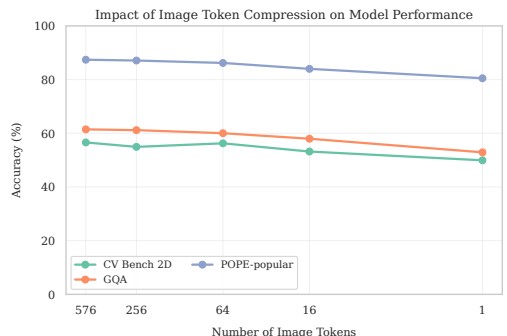

Figure 2: Performance impact of vision token compression on standard benchmarks (GQA, CV-Bench 2D, and POPE). Only minor accuracy degradation occurs, even under extreme token compression (down to a single token).

This outcome suggests two possible reasons. First, VLMs may primarily rely on vision encoder's spatial encoding, and position embedding in the LLM has low effects. Second, the current benchmarks might not effectively test the model's capacity with the LLM part of the spatial information encoding.

## 4 WHY DOES THIS HAPPEN? EMBEDDING NORM SUPPRESSION

The previous section showed a striking failure of positional encoding: permuting vision tokens often leaves predictions minor change, i.e., an unintended "bag-of-words" behavior despite the presence of positional encodings (Vaswani et al., 2017; Yuksekgonul et al., 2022). To probe *why*, we conduct an empirical norm analysis on COCO validation (5k image–text pairs) (Lin et al., 2015). We extract

vision embeddings after the MLP projector and before the language model, compute their $L_2$ norms, and compare to text-token norms. Figure 3 shows the distribution comparison.

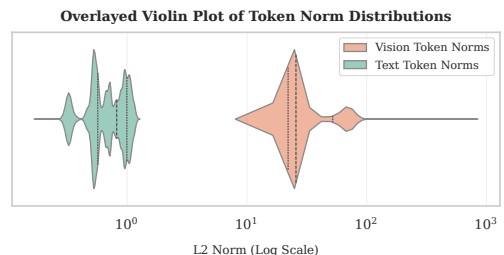

Figure 3: Distribution of $L_2$ norms for vision and text tokens in COCO validation dataset (log scale). Vision token norms range between $10^1$ and $10^3$, while text token norms range between $3 \times 10^{-1}$ and $10^0$.

In LLaVA 1.5 model, vision embeddings are typically 1–2 orders of magnitude larger than text embeddings, with an upper tail approaching 3 orders. While alignment work often emphasizes direction (cosine similarity) because logits are inner products (Hessel et al., 2021; Huh et al., 2024; Maniparambil et al., 2024), such large magnitude gaps can dominate the scaled dot-product and suppress positional signals.

Our analysis focuses on the LLaVA architecture, which uses a LLaMA decoder (Grattafiori et al., 2024). LLaMA employs a pre-Norm design where each sublayer follows a specific pattern: (1) the input *hidden states* is first normalized by RMSNorm (Zhang & Sennrich, 2019b), (2) the computational block (e.g., self-attention) processes this normalized input to produce an update, and (3) this update is added back to the original, unnormalized residual. RoPE is applied within the self-attention block to the normalized inputs (Su et al., 2024). We hypothesize that the large vision norms in the residual path are not normalized. It reduces the RoPE effectiveness applied in the attention when we consider the magnitude difference between attention outputs and unnormalized vision tokens. It might also let the vision token direction dominate due to small changes to the residual in early layers, which could cause vision attention bias.

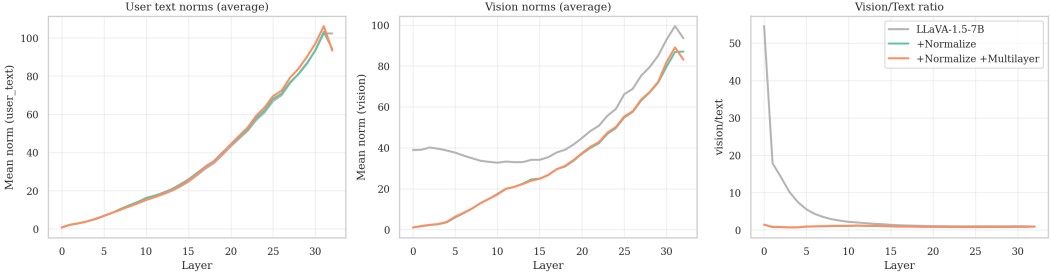

Figure 4: **Residual stream norms across depth.** Layer-wise averages of hidden state norms. Left: text tokens; middle: vision tokens; right: ratio $\frac{vision}{text}$. Vision norms are up to an order of magnitude larger than text early on, and the imbalance remains until ∼layer 15. +Normalize and +Normalize +Multilayer are our interventions in Section 6, and they balance the vision norm.

If the above mentioned hypothesis is correct, then the hidden states of vision tokens should be much larger than text tokens in early layers. On COCO val (5k), we partition tokens into vision ($V$) and user text ($T$). We exclude the fixed system prompt for clarity. We compute each token's $L_2$ norms and average over respective tokens. Figure 4 shows the results. Vision hidden states are an order of magnitude larger than text in early layers, and the disparity persists until roughly mid-depth (Figure 4 right). This directly supports the residual-scale suppression account.

More detailed explanation is given in Appendix A. Overall we want to measure the RoPE effectiveness directly to see its impact. To measure the RoPE effect directly, we can check the partial derivative of the attention w.r.t. position phase ($\phi$)'s changes. Let logits be $\ell_j = \langle q', k'_j \rangle / \sqrt{d}$ and $\alpha_k = \mathrm{softmax}(\ell)_k$ be attention weights, and partition tokens into vision ($V$) and text ($T$). Define the group masses $\alpha_V = \sum_{v \in V} \alpha_v$, $\alpha_T = \sum_{t \in T} \alpha_t$, and the group-average logit derivatives

$$g_V = \sum_{v \in V} \frac{\alpha_v}{\alpha_V} \frac{\partial \ell_v}{\partial \phi}, \qquad g_T = \sum_{t \in T} \frac{\alpha_t}{\alpha_T} \frac{\partial \ell_t}{\partial \phi}.$$

Then (derivation in Appendix A) the change of total vision attention mass w.r.t. the positional phase $\phi$ is

$$\frac{\partial \alpha_V}{\partial \phi} \;=\; \alpha_V \, \alpha_T \, (g_V - g_T). \tag{1}$$

This derivation establishes a direct link between the shift in vision attention mass ($\frac{\partial \alpha_V}{\partial \phi}$) and the underlying positional sensitivity of the vision logits ($g_V$). According to this relationship, a very small attention shift would mean low positional sensitivity. Conversely, a large attention shift would indicate high positional sensitivity, suggesting that RoPE is functioning effectively. This principle provides the mathematical support for RoPE sensitivity probe in the next section.

## 5 INTERPRETABILITY TOOLKIT: UNDERSTAND WHO IS LOOKING AT WHAT

We develop a compact toolkit to probe *where* spatial information is (or is not) used inside VLMs. The tools target three complementary questions: (i) *Does position embedding matter?* (Position Sensitivity), (ii) *Which components actually look at vision?* (Cross-Modality Balance at the head level), and (iii) *How sensitive is the model to positional phase shifts?* (RoPE Sensitivity).

### 5.1 POSITION SENSITIVITY INDEX (PSI)

As discussed in Section 3, shuffling vision tokens often leaves answers mostly unchanged, suggesting limited use of spatial order through position embedding. We quantify this with the **Position Sensitivity Index**:

$$\text{PSI} \;=\; \frac{\text{Acc(original order)} - \text{Acc(permuted order)}}{\text{Acc(original order)}}. \tag{2}$$

PSI is simple, scale-free, and interpretable: $\text{PSI} = 0$ means complete order invariance, and larger values indicate greater reliance on the token order. We report PSI at both the model and dataset levels since it could be both sides' issues. This lets us compare models and datasets on a common measure.

### 5.2 CROSS MODALITY BALANCE (CMB)

Functional behavior in Transformers is often head-specific, for example there are induction, copy, memory, summary, truthfulness attention heads in LLM (Zheng et al., 2024; Bietti et al., 2023; Tigges et al., 2023; Olsson et al., 2022; Li et al., 2023b). We therefore work at head granularity to give a closer look at the vision and text modality balance.

Let $V$ and $T$ denote the index sets of vision and text tokens, respectively. For a fixed decoding step and attention head $h$ with attention weights $a_{\cdot,h}$, we have:

$$\text{CMB}_h \;=\; \frac{\sum_{v \in V} a_{v,h}}{\sum_{i \in V \cup T} a_{i,h}}. \tag{3}$$

where $\text{CMB}_h \in [0, 1]$ measures how much attention head $h$ attends to vision versus text. In practice, we summarize the CMB with layer-by-layer head heatmaps. We can visualize which head is more focused on the vision and which is more focused on the text.

### 5.3 RoPE SENSITIVITY PROBE

To empirically quantify the practical impact of RoPE's positional information on the models' attention mechanism, we introduce a RoPE sensitivity probe to measure how much positional phase (RoPE) affects what the model predicts. For a fixed query position, we first extract the query vector and all key vectors to compute the baseline attention score, denoted as $\alpha$. We then create a perturbed state by applying an additional RoPE rotation of $\Delta$ steps exclusively to *vision keys*. The query vector and text key vectors remain unchanged and preserve original causal masking. Finally, we recompute the attention score, $\alpha_\Delta$, using the perturbed vision keys. This perturbation allows us to isolate how the positions of visual evidence shift relative to a fixed linguistic query. From this procedure,

we report two key measures that serve as a finite-difference approximation of terms in Equation 4, which is also listed here for its importance:

$$\frac{\partial \alpha_V}{\partial \phi} \;=\; \alpha_V \, \alpha_T \, \big(g_V - g_T\big). \tag{4}$$

**(A) Attention-level sensitivity.** As an approximation of $\frac{\partial \alpha_V}{\partial \phi}$, $\Delta \alpha_V$ is defined as the finite difference between $\alpha$ and $\alpha_\Delta$ with step $\Delta$. This captures the observed change in how much attention mass moves among vision tokens under a RoPE phase shift.

**(B) Intrinsic (logit-level) sensitivity.** We can also directly measure $\Delta g_V$ which is the group-average logit derivative w.r.t. the RoPE angle. It is an approximation of the $g_V$ term. In experiments we report both the attention-level difference and the normalized logit-level difference.

### 5.4 A **2D** SYNTHETIC SPATIAL BENCHMARK (2DS)

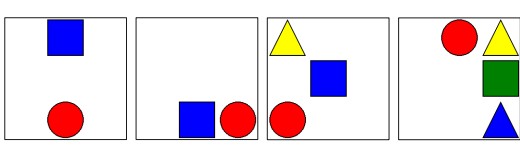

Figure 5: **2DS** examples. Left: two-object layouts; right: three/four-object layouts. Semantics are simple to focus on spatial relations.

Standard benchmarks often admit semantic shortcuts (object co-occurrence, language priors), masking spatial failure modes (Section 3 and Fig. 2). Our dataset **2DS** removes these shortcuts: scenes contain colored shapes placed at controlled locations; queries require absolute (e.g., "bottom") or relative (e.g., "left of") relations, with semantics (color/shape) serving only as identifiers. We generate multiple meta-categories by number of objects (2–6), random-ize positions within each, and enumerate questions across {color, shape, color+shape} × {absolute, relative}. Full construction details are in Appendix B. We use the same methods from Section 3 to test 2DS's position effectiveness, and the results are reported in Appendix B.2. It shows a much higher sensitivity over position compared to other datasets.

## 6 PROBING SPATIAL AWARENESS: A CONTROLLED STUDY

Guided by the diagnostics above, we test two concrete hypotheses about why VLMs underuse spatial cues under perturbations:

**H1 (Norm-induced position embedding weakness).** A modality norm mismatch (vision $\gg$ text) inflates vision token norms and attenuates RoPE effects.

**H2 (Mid-layer spatial richness).** Spatial detail is concentrated in intermediate vision-encoder layers, and final layers are more semantic, less spatial (Wang et al., 2023; Jiang et al., 2024).

### 6.1 INTERVENTIONS AND SETUP

We use LLaVA-1.5 (7B) (Liu et al., 2023) as the baseline, reproducing its two-stage training and data. We propose and evaluate the following two minimal interventions. From the derivative in equation 4, H1 predicts that reducing the vision norm can improve positional sensitivity. H2 predicts intermediate layers can add additional gains from richer spatial evidence.

**+ Normalize (H1).** RMS normalization (Zhang & Sennrich, 2019a) of vision embeddings *after* the linear projector and *before* feeding the LLM, calibrated to typical text-token norms (empirically: mean $\approx 0.83$, max $\approx 1.22$).

**+ Normalize + Multilayer (H2).** Concatenate intermediate vision features to the vision token dimension (layers 12, 16, 20, 24) following (Jiang et al., 2024), then apply the same normalization. This aims to inject fine-grained spatial signals while controlling scale. Multilayer adds a small compute overhead in theory with extra embedding expanded from 1024 to 4096 dimension. We showed empirically it is negligible in Appendix D.

### 6.2 POSITION SENSITIVITY INDEX

PSI could measure a two-fold problem – model and dataset. Table 2 compares both for PSI. If we fix one dataset, we can compare across the models, and vice versa. Given a baseline model, PSI isolates model's order use: if shuffling vision tokens barely moves accuracy compared to the baseline, the model likely underutilizes spatial arrangement. We can also compare it across benchmarks to see which one is more sensitive to the position changes. Particularly, we find 2DS has a significantly higher PSI compared to other datasets across different models, which means that 2DS can test position embedding more effectively than existing benchmarks.

Table 2: Position Sensitivity Index (%).

| Data | LLaVA 1.5 | + Normalize | + Normalize + Multilayer |
|------|-----------|-------------|--------------------------|
| VQAv2 | 1.09 | 2.03 | 2.08 |
| POPE | 0.34 | 0.35 | 0.46 |
| GQA | 4.47 | 6.10 | 5.64 |
| CV-Bench 2D | 0.57 | 11.31 | 9.72 |
| 2DS | 41.07 | 61.21 | 54.21 |

(1) **Normalization substantially increases PSI** across diverse benchmarks, consistent with our hypothesis H1: reducing norm skew restores RoPE leverage and raises order-use. (2) **Multilayer occasionally trails normalization on PSI** (e.g., CV-Bench 2D, 2DS), yet as shown in Table 3 and Appendix E it can yield higher accuracy, indicating that models can exploit spatial content from vision encoder features even if their explicit sensitivity to token permutations is moderate. In short, PSI probes order use, while accuracy reflects spatial competence.

### 6.2.1 WHICH ATTRIBUTES ARE IMPORTANT?

Table 3: 2DS accuracy (%) by attribute/relation. ↑ higher is better. Δ vs. baseline in parentheses.

| Category | LLaVA 1.5 | + Normalize | + Normalize + Multilayer |
|----------|-----------|-------------|--------------------------|
| Color_abs. ↑ | 79.60 | **88.00** (+8.40) | 87.80 (+8.20) |
| Color_rel. ↑ | 37.00 | 42.40 (+5.40) | **47.20** (+10.20) |
| Shape_abs. ↑ | 78.60 | 79.00 (+0.40) | **82.20** (+3.60) |
| Shape_rel. ↑ | 34.40 | 32.80 (-1.60) | **40.80** (+6.40) |
| Shape_color_abs. ↑ | 70.80 | 71.80 (+1.00) | **80.20** (+9.40) |
| Shape_color_rel. ↑ | 39.40 | 41.80 (+2.40) | **50.60** (+11.20) |
| Overall Acc. ↑ | 56.63 | 59.30 (+2.67) | **64.80** (+8.17) |

Table 3 showed that +Normalize chiefly helps categories where color anchors spatial relations (Color_abs./rel., Shape_color_*), suggesting that final-layer features retain strong color cues but weaker geometry. By contrast, +Normalize+Multilayer substantially lifts relative categories across the board, aligning with H2: intermediate features furnish finer geometry and local layout that compose with color cues. The largest jumps appear where both feature identity and geometry must be integrated.

On 2DS, +Normalize attains the highest PSI (61.21%) while +Normalize+Multilayer yields the highest accuracy (64.8%). This suggests that restoring positional leverage (H1) and supplying spatial content (H2) are complementary spatial information encoding. Their relative contributions differ by metric.

### 6.3 CROSS MODALITY BALANCE: WHERE DO ATTENTION HEADS LOOK?

**System prompt confound.** Prior work reports very low vision attention vs. text (Chen et al., 2024b; 2025). Our toolkit separates system prompt from user text and vision, revealing why. Text can break into two categories, the system prompt, and the user input. We compare those two and the vision attentions. As in Fig. 6, the static system prompt absorbs a large, image-invariant portion of attention (72.1%), masking vision use. It might act as an attention sink (Xiao et al., 2024). It provides a new explanation on why prior work reports very low vision attentions. When we exclude the system prompt from our analysis, a different picture emerges: in the original VLMs, the attention mass on vision tokens is significant higher than the user input text. However, we believe the apparent visual dominance is not a sign of a healthy mechanism. Instead, it is a "brute

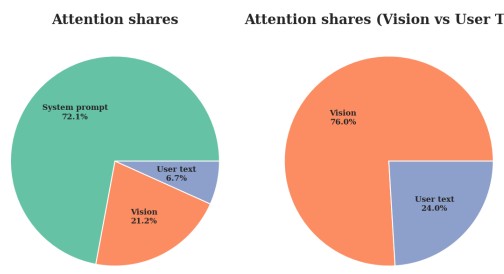

Figure 6: LLaVA 1.5 7B attention share average over COCO Validation dataset. Green is system prompt, orange is vision token, and blue is user input text. Left graph includes system prompt. Right graph just compares vision and input text.

force" consequence of the large vision token norms, as described in Section 4. The normalization can balance the vision and user input text attention as we show in Appendix F.

**Modality-integration patterns.** To understand how the VLMs integrate the vision and text modality, we visualize the CMB with heatmaps across the layers and attention heads. It can give us a granular sense of what each attention head focuses on, vision or user input text. Figure 7 showed the CMB heatmap visualization average across 5k COCO val results.

With prompt excised for clarity, LLaVA-1.5 shows many vision-specialized heads scattered across depth. It aligns with the analysis in Section 4 that the vision tokens could dominate the attention calculation. **+Normalize** exhibits a vision to text progression: early layers lean on vision, deeper layers shift toward text, indicating more balanced cross-modal exchange once norm disparity is removed. **+Normalize+Multilayer** displays a striking stage-like pattern: the first two layers emphasize vision, an early-block pivot concentrates on text, and deeper layers re-amplify vision with smooth transition. We view it as a pipeline that extract, align, then integrate. We also provide more individual image visualization and more dettails of calculation in Appendix G.

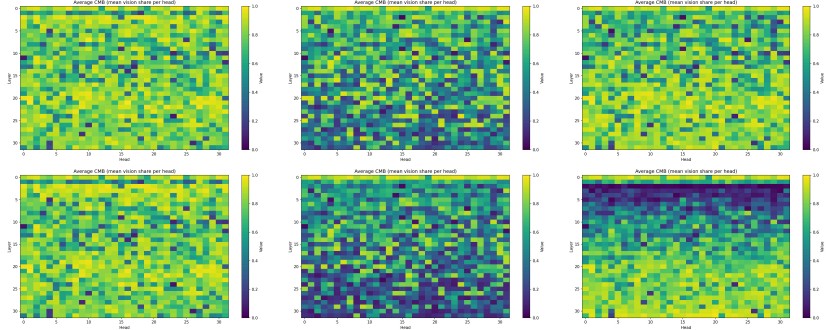

Figure 7: CMB heatmaps average (vision share) by layer×head on COCO val. Top row: inference with system prompt; Bottom row: inference without system prompt. Columns: LLaVA 1.5 7B, +Normalize, +Normalize+Multilayer. Green = vision-heavy; blue = text-heavy.

The performance of models over standard benchmarks in Appendix E show that +Normalize+Multilayer is the best, then +Normalize, then original. It suggest original model's "visual dominance" does not mean a better visual performance. With large vision activations in the residual stream (Sec. 4), the normalized state is pre-aligned to the vision subspace, so queries favor vision keys by default, and high CMB reflects a directional bias towards vision, not use question conditioned retrieval. As our attention visualizations and entropy measures indicate (Appx. H), baseline vision maps are diffuse and often focus on the wrong target objects asked in text. In contrast, +Normalize+Multilayer's staged vision-text-vision has the most focused (lowest-entropy) attention maps and the best spatial accuracy.

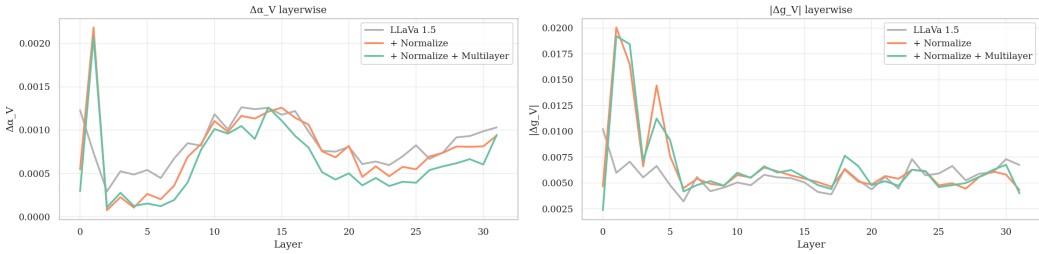

Figure 8: RoPE sensitivity (higher = larger change under phase shift) across layers.

## 6.4 ROPE SENSITIVITY: HOW DOES NORMALIZATION HELP?

One key question we want to understand is whether the normalization helped or not. We already see that the normalization can effectively balance the hidden states in Section 4 Figure 4. However,

to have a more indepth understanding, we want to know how RoPE directly changes vision. Our RoPE probe perturbs positional phase for *vision keys only* and measures (Fig. 8): (i) attention-level change $\Delta\alpha$, and (ii) the normalized logit change $\Delta g_V$, which factors out attention imbalance as we observed from CMB results. We show the RoPE probe with $\Delta = 1$ applied to the vision keys, and we report the RoPE sensitivity measures over COCO dataset average in Figure 8. We see both $\Delta\alpha_V$ and $\Delta g_V$ increase the sensitivity in early 0-5 layers over RoPE changes on vision tokens.

Both **+Normalize** and **+Normalize+Multilayer** show markedly higher sensitivity than baseline in early layers (0–5). (Fig. 8). This matches H1: reducing norm skew raises sensitivity of the RoPE effects. We believe it shows that normalization helps restoring vision token positional information from RoPE, especially in early layers. Position embedding increase the spatial channel information, improving both order-use (PSI) and task competence (2DS).

## 7 DISCUSSION AND FUTURE WORK

**What does this reveal about VLM internal mechanisms?** Spatial signals enter through (i) *vision-encoder features* and (ii) the *LLM's positional mechanism* (RoPE). Our probes indicate that both are necessary but they play different roles. Scale matching (Sec. 4) restores order sensitivity (higher PSI and RoPE probes). Exposing intermediate vision layers supplies finer geometry and local layout that raise spatial competence even when PSI changes modestly. The division of labor therefore shifts with design: **+Normalize** leans on RoPE-driven order, and let the LLM do more works on spatial information. **+Normalize+Multilayer** delegates more spatial work to the vision encoder.

**Why are raw vision attentions "low"?** Layer-averaged attention under-counts visual use because the static *system prompt* behaves like an attention sink. Once we excluded it for analysis only, many attention heads are vision-preferring (Section 6.3), especially early/mid layers. Normalization rebalanced this imbalance caused by residual norm (vision share drops from $\sim$75% to $\sim$54% as shown in Appendix F). Results suggest a more balanced vision and text could perform better, possibly due to a better question conditioned retrieval. Vision attention are also less diffused and focus less on the wrong object as shown in Appendix H.

**How does the LLM integrate modalities?** CMB heatmaps suggest distinct pipelines. The baseline scatters vision-focused heads without a clear progression. **+Normalize** shows a smoother vision→text shift with depth, matching restored positional leverage. **+Normalize+Multilayer** exhibits a stage-like vision→text→vision pattern (extract → align → integrate), with narrow, target-specific attention maps. These complementary changes explain why normalization boosts order use while multilayer boosts accuracy on relation-heavy tasks.

**Limitations and Implications** Our normalization is blunt and can reduce dynamic range, especially for 3D cues. It reduces the range of the expression and might lead to reduced 3D performance. Future work should consider a more sophisticated normalization alignment. Our work is limited to the LLaVA style model, which has a pre-trained vision encoder align with a pre-trained LLM through a projection layer. Recent VLMs train vision encoders from scratch and might not have the problem we mentioned (Bai et al., 2025; Grattafiori et al., 2024).

## 8 CONCLUSION

We show how VLMs use spatial information, and we provide a compact interpretability toolkit to make this visible and measurable. Best to our knowledge, we are the first to connect embedding–norm skew to diminished positional sensitivity in the LLM (RoPE), with analysis and experiments supporting the link. This mechanistic understanding reveals how modality-specific scales can inadvertently suppress cross-modal information flow. We introduce four simple, targeted tools for spatial information measurements: **PSI** for order use, **CMB** for head-level modality balance with prompt isolation, a **RoPE sensitivity probe** for intrinsic phase response, and **2DS**, a controlled spatial benchmark. We showed two minimal changes and clarified design levers: **Normalize** (scale-match vision to text) restores RoPE leverage and raises PSI and RoPE sensitivity; **Multilayer** adds mid-layer geometry and boosts performance accuracy. We believe those tools and insights are valuable for future VLMs design considerations.

## REPRODUCIBILITY STATEMENT

We will release code, configs, and scripts upon publication.

### DIAGNOSTICS

**Vision token permutation.**   We permute vision tokens *after* the projector (vision→LLM interface) and *before* concatenation with text, holding all other inputs fixed.

**Vision token compression.**   We insert a pooling layer *after* the projector to reduce the number of vision tokens; all other components remain unchanged.

**Embedding/hidden-state norms.**   For *embedding norms*, we pass images through the vision encoder and the projector MLP, and compute $L_2$ norms per token; text norms are computed from the LLM embedding layer. For *hidden states*, we enable `output_hidden_states=True` and record $h^{(\ell)}$. We average norms over COCO val (5k) separately for image and text caption.

**Interpretability toolkit (PSI, CMB, RoPE).**   Formulations are in Sec. 5. **CMB:** we use model-returned attention, summing over vision vs. user text indices. **RoPE sensitivity:** we hook $q, k$, rotate *vision keys only* by $\Delta$, recompute attention, and measure (i) $\Delta\alpha_V$ and (ii) $\Delta g_V$ from logits and $\alpha_v, \alpha_V$.

**2DS dataset.**   Provided as supplementary material.

### MODEL INTERVENTIONS

**+Normalize.**   We apply RMSNorm to vision embeddings *after* the projector and scale to match measured text-token norms (mean $\approx 0.83$, max $\approx 1.22$).

**+Normalize+Multilayer.**   We tap intermediate vision layers (12, 16, 20, 24) concatenating them directly. All other training details follow LLaVA-1.5 and +Normalize setting above.

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

## A    Theoretical Analysis of Large-Norm Effects on RoPE

We consider a pre-norm Transformer block (as used in LLaVA-style decoders). For token state $h^{(l)} \in \mathbb{R}^d$ at layer $l$,

$$\underbrace{a^{(l)}(\phi)}_{\text{attn out}} = \text{Attn}\big(\text{RMSNorm}(h^{(l)}), \phi\big), \qquad h^{(l+1)} = \underbrace{r^{(l)}}_{\text{residual carry}} + a^{(l)}(\phi),$$

where $\phi$ denotes the RoPE phase and $r^{(l)} = h^{(l)}$ (the pre-attention input).

Keys and queries are formed from *per-token RMS-normalized* inputs; RoPE applies a norm-preserving rotation to $q, k$ (Su et al., 2023). However, the residual is added to every normalization results even after the post layer norm. It causes the unbalanced vision token norm to keep passing on.

**Residual-scale suppression (directional lemma).** With $h^{(l+1)} = r^{(l)} + a^{(l)}(\phi)$ and $u(\phi) = h^{(l+1)}/\|h^{(l+1)}\|$. With some quotient rule and replacing $\frac{\partial \|h^{(l+1)}\|}{\partial \phi} = \frac{(h^{(l+1)})^T}{\|h^{(l+1)}\|} \frac{\partial h^{(l+1)}}{\partial \phi}$ we can have

$$\frac{\partial u}{\partial \phi} = \frac{(I - uu^\top)}{\|h^{(l+1)}\|} \frac{\partial a^{(l)}}{\partial \phi}.$$

When $\|r^{(l)}\| \gg \|a^{(l)}\|$ the magnitude of $\partial u/\partial \phi$ is suppressed by $\sim 1/\|r^{(l)}\|$ in the vision token part. Particularly, the current layer's sensitivity is proportional to the attention sensitivity of the previous layer divided by a residual norm. A large vision token norm would mean much less effective position sensitivity. One of the key terms is the attention level sensitivity. It complements attention-level analyses by explaining why even nontrivial attention shifts may yield small representational changes when the residual dominates.

### B.  Attention-level phase sensitivity (group form)

Within a given head, let logits be $\ell_j = \langle q', k_j' \rangle / \sqrt{d}$, and define attention weights $\alpha_j = \text{softmax}(\ell)_j = \frac{exp(\ell_j)}{\sum_i exp(l_i)}$. Partition keys into vision $V$ and text $T$, and define group masses

$$\alpha_V = \sum_{v \in V} \alpha_v, \qquad \alpha_T = 1 - \alpha_V.$$

Let the group-average logit derivatives be

$$g_V = \sum_{v \in V} \frac{\alpha_v}{\alpha_V} \frac{\partial \ell_v}{\partial \phi}, \qquad g_T = \sum_{t \in T} \frac{\alpha_t}{\alpha_T} \frac{\partial \ell_t}{\partial \phi}.$$

A standard calculation with some quotient rule from softmax yields the single vision key given a fixed query:

$$\frac{\partial \alpha_v}{\partial \phi} = \alpha_v \left( \frac{\partial \ell_v}{\partial \phi} - \sum_k \alpha_k \frac{\partial \ell_k}{\partial \phi} \right) \tag{5}$$

We can use the above notation and get the group-level sensitivity by aggregating all vision attentions then substitute with $\alpha_V, \alpha_T, g_V, g_T$

$$\frac{\partial \alpha_V}{\partial \phi} = \alpha_V \left( g_V - \sum_k \alpha_k \frac{\partial \ell_k}{\partial \phi} \right) \tag{6}$$

$$= \alpha_V \left( g_V - \left( \sum_{v \in V} \alpha_v \frac{\partial \ell_v}{\partial \phi} + \sum_{t \in T} \alpha_t \frac{\partial \ell_t}{\partial \phi} \right) \right) \tag{7}$$

$$= \alpha_V \left( g_V - (\alpha_V g_V + \alpha_T g_T) \right) \tag{8}$$

$$= \alpha_V \alpha_T \left( g_V - g_T \right) \tag{9}$$

$$\tag{10}$$

Eq. equation 10 is an *attention-level* statement, and we can directly measure the change without the need to consider the previous layer.

**Small-phase probe.** If we rotate *only vision keys* by a small phase $\Delta$ (queries fixed), then $g_T$ is unchanged and to first order

$$\Delta\alpha_V \approx \alpha_V(1-\alpha_V)\,\Delta g_V,\tag{11}$$

This factorization underlies our probe: $\Delta\alpha_V$ grows when (i) the *balance factor* $\alpha_V(1-\alpha_V)$ is large (less saturation) and/or (ii) the *intrinsic* phase response $\Delta g_V$ is large.

Empirically we can capture $q, k$ inside attention and conduct position perturbation, so we can measure $\Delta\alpha_V$ and $\Delta g_V$ with attention-level sensitivity. The *effective* positional change is modulated by the residual. Reducing the interface scale (our `+Normalize`) decreases $\|r^{(\ell)}\|$ for vision tokens in early layers, mitigating residual-scale suppression and making attention-level phase shifts consequential for subsequent computation. It also confirmed with our observation that `+Normalize` increases early layers' position sensitivity in Section 6.4.

## B    2DS DATASET DETAILS

We constructed a synthetic dataset that: (1) Ensures spatial cues are essential for correctly answering queries. (2) Isolates and systematically controls semantic aspects, enabling analysis of their contributions to spatial understanding. The dataset combines semantic attributes (color, shape, or both) with spatial properties (relative or absolute positions). We denote the **2D S**ynthetic Dataset as **2DS**

**Example Queries** including:

- "What color is at the bottom of the image?" (Color, absolute position)
- "Is the circle below the square?" (Shape, relative position)

These queries illustrate that object attributes (color, shape) serve primarily as identifiers, with positional relationships being crucial for correct responses.

### B.1    DATASET CONSTRUCTION DETAILS

We created meta-categories containing images with 2, 3, 4, 5, and 6 objects. Within each category, we employed the same set of objects while randomly varying their *spatial positions*. Each meta-category comprises 100 images, resulting in a total of 500 images. For every image, we systematically asked questions spanning combinations of semantic (color, shape, color and shape) *and* spatial (absolute, relative) properties, yielding 3000 total questions. Note the emphasis is on spatial relationship reasoning.

### B.2    HOW EFFECTIVE CAN 2DS MEASURE THE POSITION EMBEDDING SENSITIVITY

As the preliminary experiments in Section 3 showed that most of the data retain the performance even with random permutation, we can use the same experiments to understand the 2DS results.

**Permutation**    Table 4 show the 2DS with permutation of the input tokens. We also report the difference which is used to compute the PSI in Table 2. We can see the +Normalized version is more susceptible to the token permutation. As the baseline, we also test with the Vicuna-7B to show the model performance is well above the random guess.

Table 4: For 2DS, we report accuracy on original (Orig.) and permuted (Perm.) token arrangements with difference (Diff.).

| Model | 2DS (%) | | |
|---|---|---|---|
| | Orig. | Perm. | Diff. |
| Vicuna-7B | 15.97 | – | – |
| LLaVA 1.5 | 56.63 | 33.37 | -23.26 |
| + Norm | 59.30 | 23.00 | **-36.30** |
| + Norm + Multi | 64.80 | 29.67 | -35.13 |

**Compression**    Another method to test the 2DS is through compression, which eventually reduce position embeddings effectiveness. If the vision encoder can successfully carry over most of the spatial information, then the compression would not hurt performance significantly. However, if the position embedding is crucial, then there should be a significant decrease. We show the exact same plot but with 2DS through the compression in Figure 9.

Here we see a significantly more drop in the 2DS performance. We believe those two measures suggest that position embedding has an important role for the 2DS performance.

## C    TRAINING DETAILS

We follow the training recipe described in LLaVA 1.5 7B (Liu et al., 2023), consisting of two stages: pretraining and instruction tuning. All experiments utilize 8 Nvidia A100 GPUs. Learning rates for

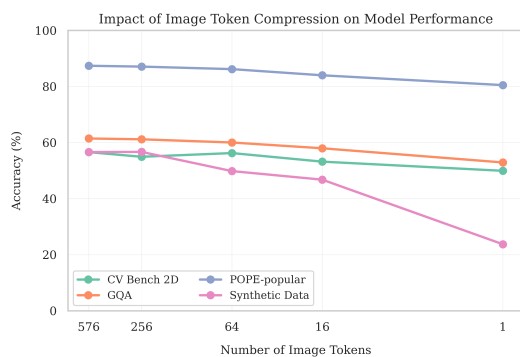

Figure 9: Performance impact of vision token compression on standard benchmarks (GQA, CV-Bench 2D, and POPE). Only minor accuracy degradation occurs, even under extreme token compression (down to a single token).

each training stage are detailed in 5, with all other hyperparameters and datasets identical to (Liu et al., 2023).

Table 5: Learning rate of training

| Model | Pre-train | Finetune |
|---|---|---|
| LLaVA 1.5 7B | 1e-3 | 2e-5 |
| +Normalize | 1e-3 | 2e-5 |
| +Normalize+Multilayer | 1e-3 | 2e-5 |

## D  ANALYSIS OF COMPUTATIONAL COST AND INFERENCE TIME

In this section, we analyze the computational overhead of our two proposed methods to evaluate their practical efficiency.

**Method 1: +Normalize.** Our first method, which applies L2 normalization to the final-layer vision tokens, introduces zero additional parameters or floating-point operations (FLOPs) during inference. The normalization is a lightweight mathematical operation on the existing embedding vector and its computational cost is negligible.

**Method 2: +Normalize+Multilayer.** Our second method enriches the visual representation by concatenating features from the intermediate and final layers of the vision encoder. This modification affects the initial visual projection layer. In the original LLaVA, the projector is an MLP that maps final 1024-dimensional feature vector to 4096-dimensional LLM embedding space. While our proposed method concatenates four 1024-dimensional feature vectors and inputs them to the projector, producing 4096-dimensional vector. While this moderately increases the number of parameters in this single projection layer, this component represents a very small fraction of the total parameters in the multi-billion parameter VLM.

**Empirical Inference Time.** To assess the real-world impact on performance, we benchmarked the inference speed of the baseline model against our two proposed methods. The experiment was conducted on a single NVIDIA RTX 4090 GPU, measuring the average time to process 500 samples from the GQA dataset validation split. All three models are within the same inference time ($0.137s \pm 0.002s)/sample$, empirically there are trivial compute cost differences.

# E  BENCHMARK PERFORMANCE

## E.1  2DS PERFORMANCE

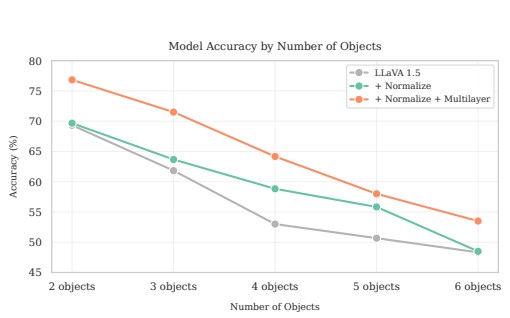

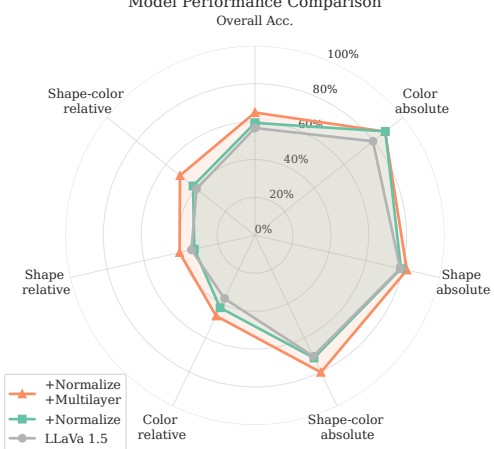

(a) Accuracy comparison across varying numbers of objects. Our interpretability-informed adjustments yield consistent improvements, especially as spatial complexity increases.

(b) Accuracy comparison across different categories of 2DS dataset. We see consistent improvements across categories.

Figure 10: Model performance across increase of objects and different sub-categories

Table 3 summarizes accuracy across distinct spatial query categories, and Figure 10b illustrates performance across varying numbers of objects, clearly highlighting the effectiveness of our interventions.

**Vision Embedding Normalization:** Normalization alone improves overall accuracy from 56.63% to 59.30%, strongly supporting our interpretability insight that balancing embedding norms significantly enhances spatial information utilization. Notably, improvements are particularly pronounced in color-related spatial relationship queries.

**Normalization + Intermediate-layer Features:** This approach achieves the highest overall accuracy (64.80%, +8.17% improvement over baseline), underscoring the importance of explicitly incorporating local spatial details. This method substantially enhances accuracy, especially in shape and color combined relative position queries.

Both methods consistently outperform the original LLaVA 1.5 model, particularly as spatial complexity (the number of objects) increases (Figure 10a), validating our hypothesis about the critical roles of embedding normalization and local spatial detail preservation.

## E.2  RESULTS ON STANDARD BENCHMARKS

Table 6: Performance (accuracy %) on standard vision-language benchmarks. ↑ indicates higher is better. Values in parentheses show difference from LLaVA 1.5 baseline.

| Category | LLaVA 1.5 | + Normalize | + Normalize + Multilayer |
|---|---|---|---|
| VQAv2 ↑ | 78.20 | 78.76 (+0.56) | **79.17** (+0.97) |
| POPE ↑ | 87.30 | 87.30 (0.00) | **87.70** (+0.40) |
| GQA ↑ | 61.46 | 62.04 (+0.58) | **62.52** (+1.05) |
| CV-Bench2D ↑ | 56.59 | **59.91** (+3.32) | 58.69 (+2.10) |

To assess generalization, we evaluate our methods on established benchmarks (Table 6) that are used in recent works (Liu et al., 2023; Tong et al., 2024a; Zhu et al., 2024; Liu et al., 2024).

Both interpretability-informed interventions yield moderate yet consistent improvements. Normalization and Multilayer Normalization consistently achieves small but meaningful gains (up to

Table 7: Performance comparison on CV-Bench 2D benchmarks. We report accuracy on original (Orig.) and permuted (Perm.) token arrangements and PSI for each of the three tasks: overall 2D spatial reasoning, counting, and relation. We also report evaluation using Vicuna-7B on top without image input but just text input as the random baseline.

| Model | CV-Bench 2D (%) | | | | | | | | |
|---|---|---|---|---|---|---|---|---|---|
| | Overall | | | Count | | | Relation | | |
| | Orig. | Perm. | PSI | Orig. | Perm. | PSI | Orig. | Perm. | PSI |
| Vicuna-7B | 31.36 | – | – | 12.31 | – | – | 54.46 | – | – |
| LLaVA 1.5 | 56.59 | 56.26 | 0.58 | 52.07 | 52.37 | -0.58 | 62.05 | 60.97 | 1.74 |
| + Norm | 59.91 | 53.13 | 11.32 | 52.81 | 51.14 | 3.16 | **68.51** | 55.54 | 18.93 |
| + Norm + Multi | 58.69 | 52.99 | 9.71 | 52.08 | 49.49 | 4.97 | 66.72 | 57.23 | 14.22 |

3.32%), confirming that improved spatial representation benefits overall reasoning without sacrificing semantic capabilities. Notably, normalization shows more pronounced gains in the CV-Bench 2D dataset, particularly for spatial relationship tasks.

Specifically, we investigate the subcategories of the CV-Bench2D as shown in Table 7. We are curious why the permutation drop was not significant. As shown in Table 7, we can check the PSI index to see whether the model is sensitive to that task, and Count is not really a spatial task since there are marginal effects when changing spatial information. In comparison, we see the Relation task is more spatial, and the +Normalize has a much better performance in that category.

# F SYSTEM PROMPT ATTENTION

We visualize the attention portion when inference with and without system prompt for all 3 models in Figure 11. We observe that the system prompt absorbs the most attention shares. Even when we exclude the system prompt during the calculation, it might skew the probability mass and changes the distribution of a much smaller vision and user input text attentions. Therefore we also demonstrate vision and user input text attention inferenced without the system prompt on the right column of the Figure 11.

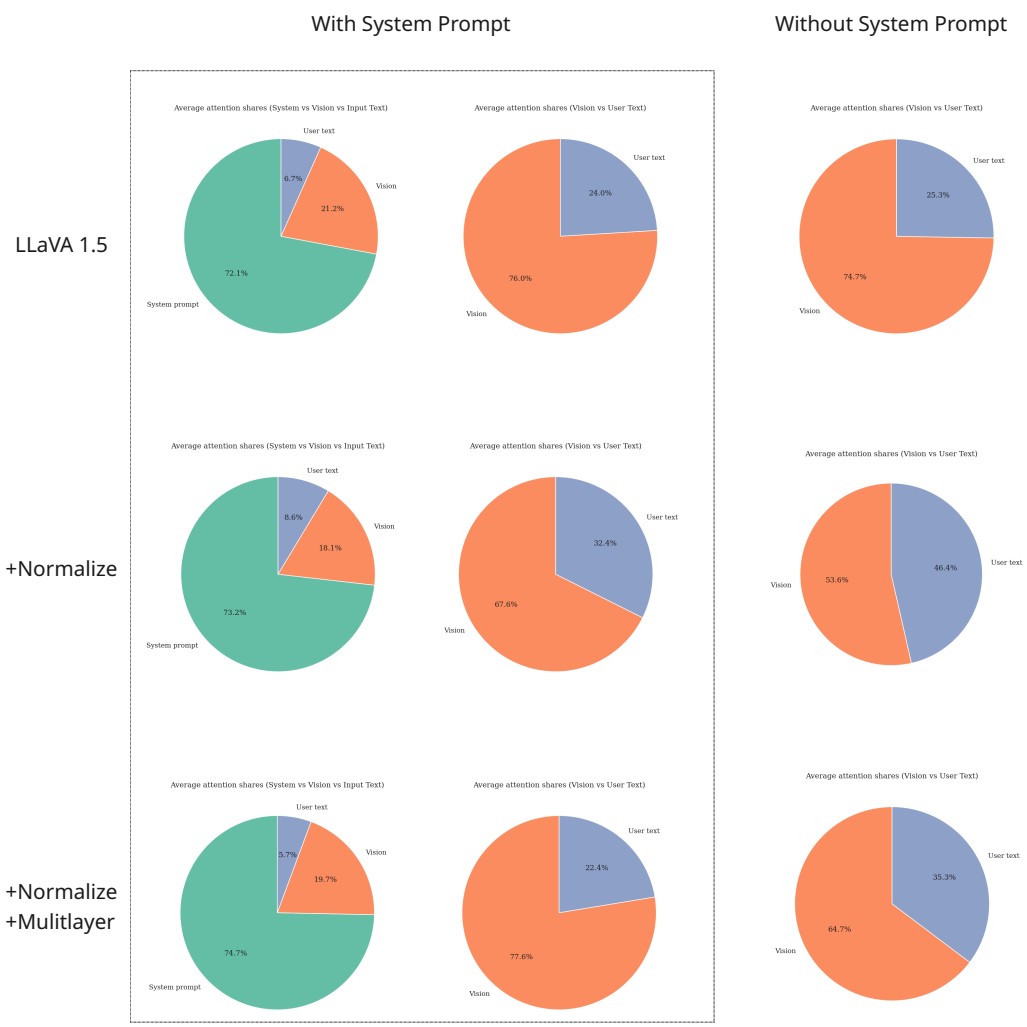

Figure 11: Visualization of system prompt, vision, and user input text attention shares. Left 2 columns are inferenced with the system prompt, then calculate the portion of the vision and user input text. Right column is inferenced without the system prompt. We see a much more balanced vision and user input text attention share in the normalized and multilayer version.

## G    ATTENTION HEAD VISUALIZATION

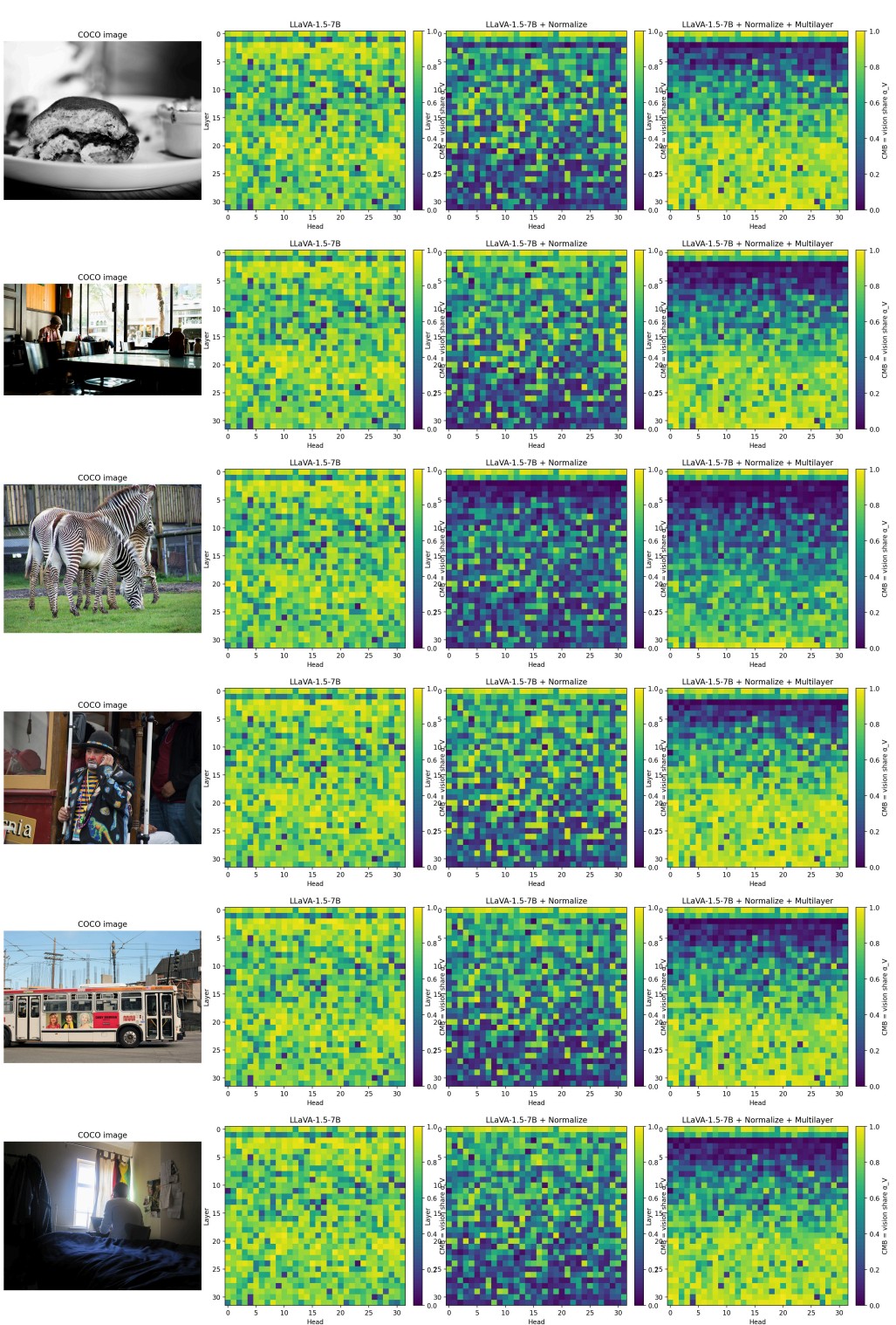

Figure 12: Visualization of each image's attention head over layers. We inference without the system prompt. CMB directly calculated from vision and text attention.

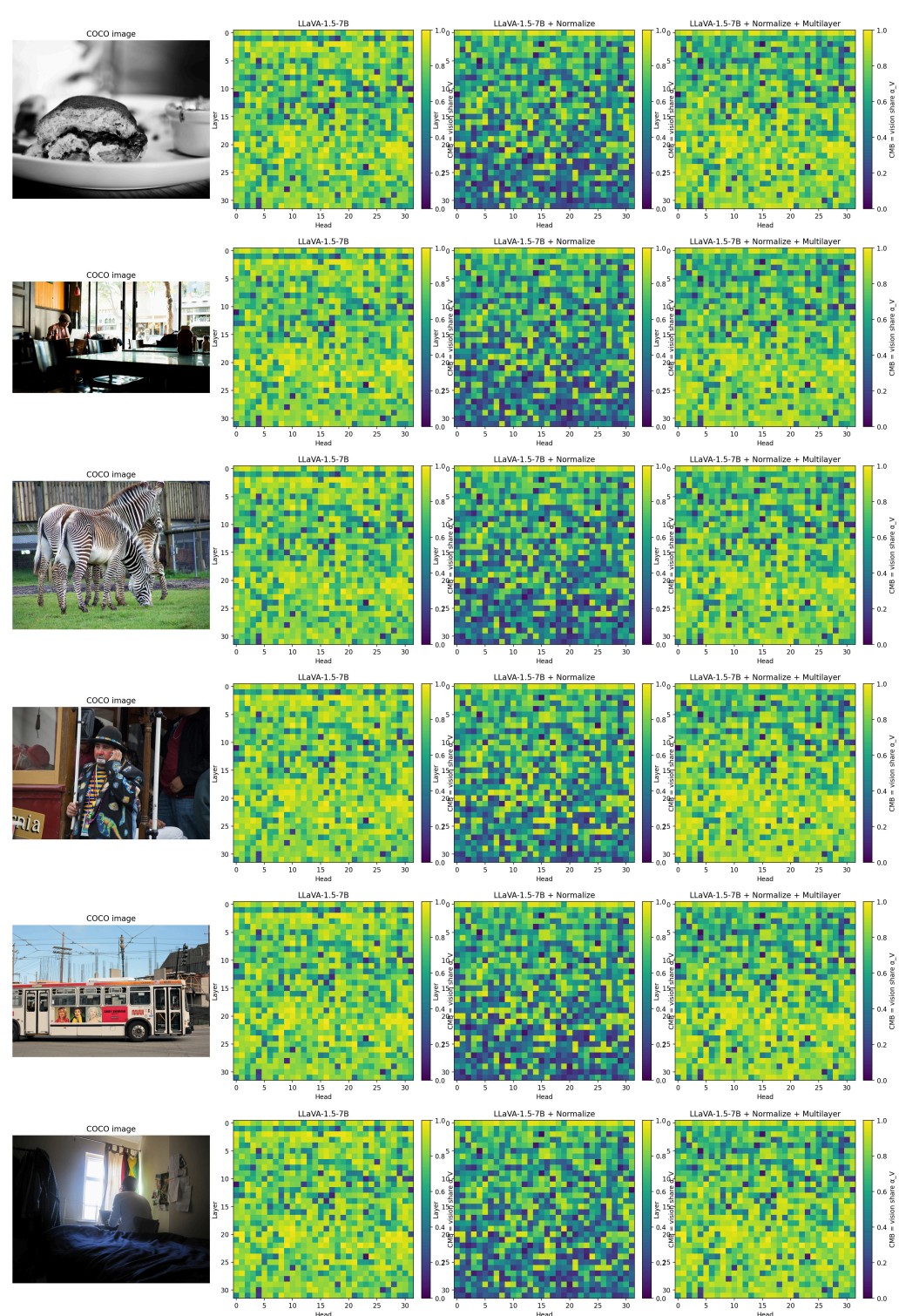

Figure 13: Visualization of each image's attention head over layers. We inference with the system prompt. CMB take values from vision and text attentions and normalize them with the sum of two.

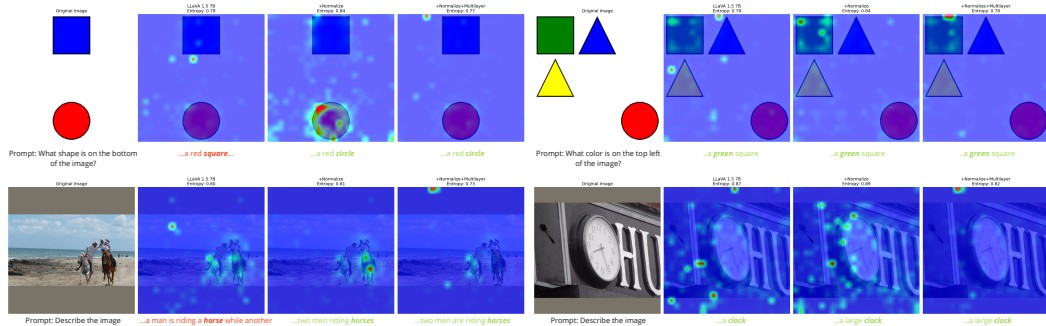

Figure 14: Visualization of self-attention patterns. We overlay the attention map on top of the image. The question for the model is under each row. We use the first target word of the response for attention map, for example the attention map is based on 'square' and 'circle' for top left rows. Entropy values are on top of each image. Red answers are wrong. Green answers are correct. Each image has entropy on top: lower means more concentrated higher means more diffused

## H  ATTENTION VISUALIZATION

We visualized the self-attention patterns from LLM to interpret the effects of each modification on spatial reasoning. Figure 14 provides representative attention maps across four illustrative queries. Each query compares three scenarios: the baseline (Original LLaVA 1.5), normalized embeddings, and normalized embeddings combined with multilayer features. We visualized the first text token of the target object in the answer, which are bold and italic words in Figure 14. More examples are in H.2.

**Baseline (Original LLaVA 1.5):** The baseline model shows diffuse and scattered attention patterns, possibly influenced by large-magnitude vision embeddings that overshadow positional cues. This diffuseness makes it difficult for the model to clearly focus on relevant spatial tokens, leading to suboptimal spatial reasoning.

**Embedding Normalization:** Embedding normalization leads to distinct and focused attention distributions, significantly increasing attention on relevant spatial tokens. Interestingly, this variant exhibits broader coverage of the attention map, suggesting that the model actively utilizes positional embeddings to locate informative spatial regions. The improved spatial focus directly contributes to enhanced performance in spatial reasoning tasks.

**Embedding Normalization + Multilayer Features:** Interestingly, the multilayer feature variant produces sparser, more selective attention distributions with lower attention magnitudes, similar visually to the original model but fundamentally different in effectiveness. Despite lower peak attention, this variant consistently performs better across most spatial tasks. We hypothesize that intermediate-layer vision embeddings inherently carry sufficient local spatial information, thus significantly reducing the need for explicit positional encoding at the LLM stage. Consequently, the model selectively focuses attention on regions already identified as informative by intermediate-layer embeddings without the need to check other regions.

### H.1  INTERPRETING ATTENTION PATTERNS AND PERFORMANCE

To quantitatively interpret these observations, we computed the normalized attention entropy (Table 8). We average all text tokens' attention over the vision tokens to compute the entropy. Lower entropy indicates more targeted attention distributions, and high scores mean more diffused attention distributions. More examples are in H.2.

The normalized embeddings exhibit higher entropy compared to the baseline, reflecting a broader and more uniform exploration of spatial regions, yet with a clear focus on the target object. This suggests that embedding normalization encourages the LLM to explicitly explore multiple spatial positions to extract positional information. In contrast, the multilayer normalized variant demonstrates the lowest entropy, underscoring the model's confidence in selectively attending to a few pre-

Table 8: Average attention entropy over different datasets. Lower entropy indicates more selective attention.

| Vision Token Attention Entropy | LLaVA 1.5 | + Normalize | + Normalize + Multilayer |
|---|---|---|---|
| 2DS ↓ | 0.76 | 0.90 | **0.72** |
| COCO Val ↓ | 0.81 | 0.89 | **0.72** |
| CV-Bench2D ↓ | 0.80 | 0.90 | **0.72** |

cise spatial regions. This observation supports our hypothesis that intermediate-layer embeddings inherently encode robust spatial details, enabling the model to rely less on explicit attention-based positional signals to look at vision tokens. In addition, it suggests that the improvements of the Normalize and Multilayer Normalize are two different mechanisms: the former is based on LLM attention, and the later is based on vision encoder spatial information.

Providing early-layer features from vision encoders potentially offers a shortcut for LLMs to directly leverage spatial cues without extensive positional encoding training. Future research should explore strategies to optimally balance intermediate-layer spatial detail and positional encodings.

## H.2 ATTENTION ENTROPY

We adapt the visualization method from (Zhang, 2024) to analyze the LLM attention specifically on vision tokens. Attention maps correspond to particular, meaningful text tokens, as labeled above each image in 15. This is a more refined version compared to average text tokens. Notably, we observe consistently lower entropy in the multilayer-normalized models.

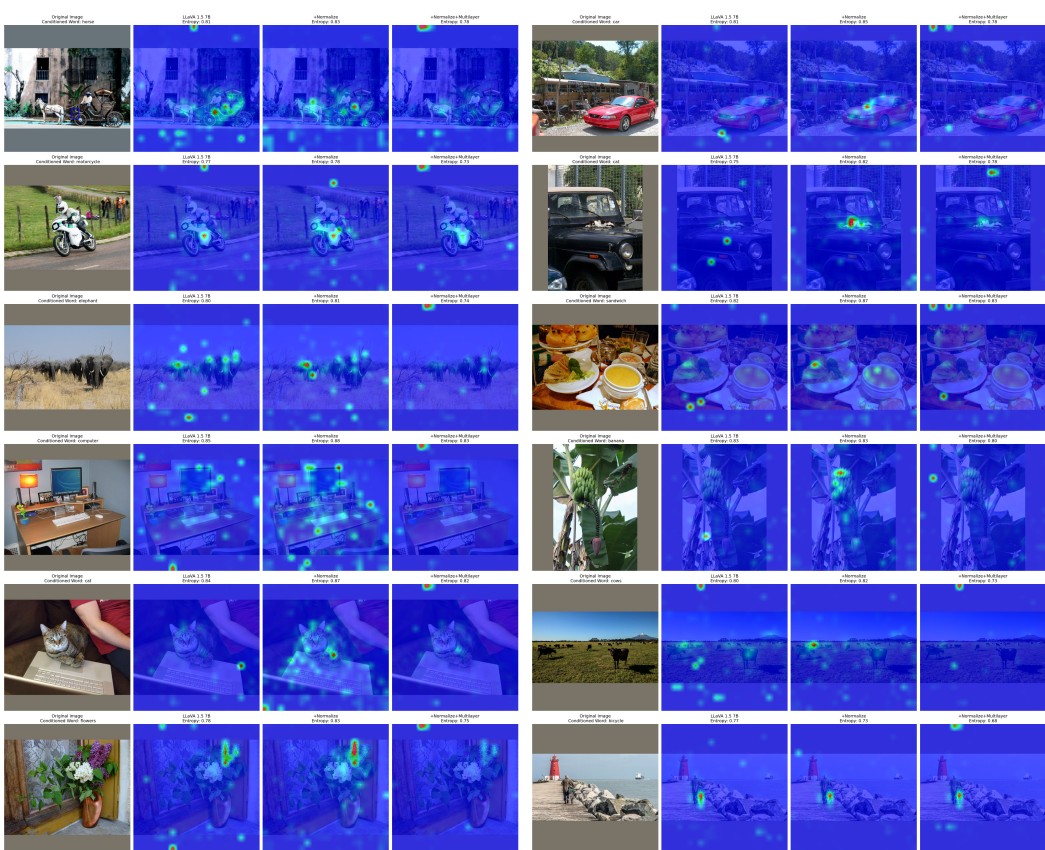

Figure 15: More visualizations

## THE USE OF LARGE LANGUAGE MODELS (LLMS)

We use LLM to polish writing and help with proofs. In addition, we use LLM to assist coding and visualization.

