# OpenReview forum: "Beyond Semantics: Rediscovering Spatial Awareness in Vision-Language Models"
_ICLR.cc/2026/Conference — Submitted to ICLR 2026_

### Official Review · Reviewer_EGGm · 2025-11-03

**Soundness:** 3
**Presentation:** 3
**Contribution:** 2
**Rating:** 4
**Confidence:** 3

**Summary:**

Vision–Language Models (VLMs) excel at identifying and describing objects but often fail at spatial reasoning. We study why VLMs, such as LLaVA, under-utilize spatial cues despite having positional encodings and spatially rich vision-encoder features. Our analysis reveals a key imbalance: vision token embeddings have much larger norms than text tokens, suppressing LLM's position embedding. To expose this mechanism, we developed three interpretability tools: (1) the Position Sensitivity Index, which quantifies reliance on token order, (2) the Cross-Modality Balance, which reveals attention head allocation patterns, and (3) a RoPE Sensitivity probe, which measures dependence on rotary positional embeddings. These tools uncover that vision tokens and system prompts dominate attention. We validated our mechanistic understanding through targeted interventions that predictably restore positional sensitivity. These findings reveal previously unknown failure modes in multimodal attention and demonstrate how interpretability analysis can guide principled improvements. We will release code upon publication.

**Strengths:**

1) The paper has good presentations.
2) Enough experiments show the effectiveness of the proposed method.
3) The topic is interesting.

**Weaknesses:**

1) Underutilization of Spatial Cues: VLMs like LLaVA fail to effectively leverage spatial information despite having access to positional encodings, leading to deficiencies in spatial reasoning tasks.
2) Imbalance in Token Norms: The significant difference in norms between vision token embeddings and text tokens suppresses the model's ability to utilize position embeddings, hindering overall performance.
3) Dominance of Vision Tokens: The attention mechanism is dominated by vision tokens and system prompts, which can overshadow spatial cues and limit the model's reasoning capabilities.

**Questions:**

Underutilization of Spatial Cues: What specific strategies can be implemented to enhance the utilization of spatial information in VLMs, ensuring they effectively leverage positional encodings for improved spatial reasoning?

Imbalance in Token Norms: How can we adjust the norms of vision token embeddings relative to text tokens to create a more balanced representation that allows for better utilization of position embeddings?

Dominance of Vision Tokens: What modifications can be made to the attention mechanism to ensure a more equitable distribution of focus between vision tokens, system prompts, and spatial cues, thereby improving overall reasoning capabilities?

---

> ### Author Response · Authors · 2025-11-22
>
> Thank you for the review. Below we address your points and answer directly. We believe the core questions are already answered in the paper. We will restate each question and what we did in the paper.
>
> >Underutilization of Spatial Cues
>
> In the paper we propose and empirically evaluate two concrete strategies to enhance spatial cue utilization, Normalize and Normalize+Multilayer. Together, these show that we can enhance RoPE by fixing norm imbalance. In addition, feeding spatially richer features can also increase spatial cue usage in current VLMs. We hope those two simple strategies could inspire more sophisticated designs for VLMs.
>
> >Imbalance in Token Norms
>
> For imbalance in token norms, our adjustment strategy is straightforward and intentionally simple for diagnosis. We just normalize them into the similar range of the text embedding norms. We measure the typical L2 norm of text embeddings then apply a RMS-style normalization to vision tokens so that their norms are brought to the same scale as text tokens.
>
> >Dominance of Vision Tokens
>
> For dominance of vision shares in attention head, we find that normalizing vision token norms reduces disproportional vision share. We believe it provides the strategies for future direction. One way is to normalize them before feeding into LLM so we can reduce the dominance of vision tokens directly, and another potential solution is to adjust the attention block and add RoPE after normalization. However, adjusting architecture could be more difficult. In this paper we focused on the simplest proof-of-concept modifications through directly normalizing the vision tokens.

---

### Official Review · Reviewer_F3a2 · 2025-11-06

**Soundness:** 2
**Presentation:** 3
**Contribution:** 2
**Rating:** 2
**Confidence:** 4

**Summary:**

This paper studies why LLaVA-1.5 underutilizes spatial cues despite having positional encodings and spatially rich vision-encoder features. It revealed that vision token embeddings have much larger norms than text tokens, suppressing LLM's position embedding. The authors proposed three interpretability tools that uncover that vision tokens and system prompts dominate attention, which reveal previously unknown failure modes in multimodal attention.

**Strengths:**

S1: This paper is well-written with a nice flow.

S2: The investigated problem of spatial awareness is very important in VLMs.

S3: The findings are very interesting (if they are general in VLMs), especially vision token embeddings have much larger norms than text tokens, suppressing LLM's position embedding.

**Weaknesses:**

W1: This study is only based on LLaVa-1.5, which is considered relatively weak and old, given the rapid growth in VLM research. The observed problems on this particular model might not generalize to other VLMs. For example, Qwen3-VL might have alleviated the spatial awareness issue, and LLaVa-1.5 may simply be less well aligned in visual and textual spaces. Therefore, my main concern is that the findings in this work could not guide further development of VLMs

W2: A very related concern is that this work only focuses on the RoPE positional encoding, which has many extensions that are widely used in more recent LLMs/VLMs. So it’s hard to justify the significance of these findings. Is it possible to cover other VLMs with different positional encoding strategies?

W3: Since this paper includes a discussion of cross-modality balance (Section 5.2), it would be beneficial to include related works about modality imbalance issues in VLMs, including but not limited to IsoBench (COLM ‘24) and SEAM (COLM ‘25).

W4: The examples and constructed benchmark mainly focus on recognition rather than reasoning. I am concerned that in those more complex visual reasoning tasks, for example, Chess boards as image inputs, the permutation will result in very different outputs. However, the limitation on recognition makes the task too easy, so that the reliance on the image token ordering becomes less important. Is it possible to see if the conclusions still hold for visual reasoning tasks?

W5: VLMs are trained with the unnormalized distribution of token embeddings of the last layer. I don’t understand how it is possible that the model can still behave normally out-of-the-box when normalization and multi-layer features are incorporated without further alignment. It will be great if the authors could explain in more detail.

W6: The authors promised to release code upon publication, but the reproducibility is limited at the current stage. Any specific reason for this plan?

**Questions:**

Please refer to W2 and W4-6

---

> ### Author Response · Authors · 2025-11-22
>
> Thank you for the detailed and thoughtful comments. We address each concern below.
> >This study is only based on LLaVa-1.5
>
> For the generalization to newer VLMs, our goal is to understand and improve the design pattern of coupling a pre-trained vision encoder with a pre-trained LLM using relatively limited vision–language  alignment data. There are many vision data, and many language data, but vision and language overlap data is much less. We believe aligning the pre-trained vision encoder with a pre-trained LLM is important.
> Many recent high-performing VLMs (e.g., Qwen2.5-VL and possible Qwen3-VL) adopt a different design choice: they “train the redesigned ViT from scratch” for the VL setting [1]. This approach might have a better alignment but substantially changes the regime we study.
>
> To probe generality within the pretrained-vision + pretrained-LLM design, we additionally analyzed DeepSeek-VL2 [2], which also uses a pre-trained vision encoder and LLM but is trained on much larger data, estimated to be x410 times larger than LLaVA-1.5. For DeepSeek-VL2-tiny and -small, we observe the same qualitative pattern: vision token norms still remain systematically larger than text token norms, although the gap is reduced compared to LLaVA-1.5 (e.g., average vision/text norm ratios ≈1.5–1.8 vs. much larger ratios in LLaVA-1.5). DeepSeek-VL2’s training uses roughly 1.2M pairs just for alignment (similar to LLaVA-1.5’s entire training), plus 798.5B tokens for pretraining and 19.5B for SFT—over two orders of magnitude more data overall. This enormous scale of training data appears to partially reduce norm mismatch, but at a very high data/compute cost.
>
> | Model                          | Text Mean | Text Std | Vision Mean | Vision Std | Ratio Mean | Ratio Std |
> |--------------------------------|----------:|---------:|------------:|-----------:|-----------:|----------:|
> | deepseek-ai/deepseek-vl2-tiny |     5.88  |    2.45  |       9.09  |      5.71  |      1.55  |     2.33  |
> | deepseek-vl2-small            |     3.98  |    1.13  |       7.26  |      5.04  |      1.82  |     4.45  |
>
>
> Our work shows that norm imbalance remains a robust phenomenon even in more powerful VLMs.
>
> > this work only focuses on the RoPE positional encoding
>
> We agree that many modern LLMs/VLMs extend or modify RoPE. Importantly, our claim is not that RoPE itself is flawed, but that large cross-modality norm imbalances during vision–language alignment can suppress any positional signal whose magnitude is small relative to the content embedding. Most positional encoding variants, including RoPE extensions, do not significantly change embedding norms by design—they modulate phase or direction rather than overall magnitude. Different position embedding does not normalize the large vision norms

---

> > ### Author Response · Authors · 2025-11-22
> >
> > >include related works
> >
> > We appreciate the pointer to related work on modality imbalance. Section 5.2 currently discusses cross-modality balance conceptually, but we agree that explicitly connecting to recent systematic studies would strengthen the paper. We will add and discuss works
> >
> > >The examples and constructed benchmark mainly focus on recognition rather than reasoning
> >
> > For the visual reasoning tasks, we believe there is an important distinction between visual recognition and visual reasoning. Our current scope focuses on spatial recognition and localization, and we see this as a necessary foundation for more complex reasoning. Many visual reasoning pipelines (e.g., using tools to crop regions, track objects, or zooming in) implicitly depend on accurate recognition of coordinates and spatial relations. If the model is insensitive to token ordering in recognition-style tasks, this can undermine downstream reasoning. In the chess board example you give, we believe that if a model cannot differentiate the spatial locations well, then the subsequent visual reasoning would also be affected.
> >
> > >how it is possible that the model can still behave normally out-of-the-box
> >
> > For ‘out-of-the-box’ clarification, we do not apply Normalize or Multilayer as a post-hoc patch on a fully trained model. Instead, as stated in Section 6.1, we retrain the LLaVA-style alignment from scratch using the same two-stage pipeline, data, and hyperparameters as the original LLaVA-1.5, but with our modified designs. We will clarify this.
> >
> > >release code upon publication
> >
> > For reproducibility and code release plan, we have training pipelines, evaluation scripts, models, and dataset prepared and ready to be released.
> >
> > [1] Bai, Shuai, et al. "Qwen2. 5-vl technical report." arXiv preprint arXiv:2502.13923 (2025).
> >
> > [2] Wu, Zhiyu, et al. "Deepseek-vl2: Mixture-of-experts vision-language models for advanced multimodal understanding." arXiv preprint arXiv:2412.10302 (2024).

---

### Official Review · Reviewer_rZyW · 2025-11-08

**Soundness:** 3
**Presentation:** 3
**Contribution:** 3
**Rating:** 2
**Confidence:** 3

**Summary:**

The paper studies why VLMs underuse spatial and positional cues. They introduce metrics like PSI, CMB, and RoPE-sensitivity to measure position use, and apply two light interventions, Normalize and Normalize+Multilayer, and reports consistent gains, both on interpretability metrics and spatial reasoning benchmarks.

**Strengths:**

1. The paper makes the "underuse of spatial cues" concrete. The finding that perturbing the RoPE presentation barely hurts performance is a strong and clear signal of the problem.
2. The interventions are minimal and simple to implement.
3. RoPE-sensitivity is a good probe than pure attention scores. It gives a more faithful read of whether the model actually uses positional information.

**Weaknesses:**

1. The experiment (compress to extreme)  shows in Figure 2 is confounded. It changes token count, spatial frequency bandwidth, and aggregation at the same time. This mixes the effect of "LLM’s RoPE only" with "extreme compression", not strong enough support the motivation presented in this section.
2. The link between Normalize and RoPE is indirect. The claimed mechanism is "Normalize balances vision tokens and RoPE embedding", which is not so direct to make the model sensitive to RoPE embedding. It also highlights text tokens. Besides, "+Multilayer" likely improves by adding richer features (an augmentation effect), which is not aligned with the stated goal of emphasizing position or highlighting RoPE.
3. Normalize is treated as the key factor, but there is no continuous control study (for example, gradually scaling visual/text norms, or selectively scaling subsets of tokens) to establish a causal dose-response curve.
4. VQAv2/GQA are general or compositional VQA, not designed for left/right/front/back/orientation/reference frame skills. POPE measures hallucination, which is orthogonal to spatial reasoning. The authors could consider adding What’sUp or GSR-Bench to directly test spatial relations.
5. The symbols in the figures are too small, making them hard to read (e.g., in Figure 7).

**Questions:**

Please refer to Weaknesses.

---

> ### Author Response · Authors · 2025-11-22
>
> We appreciate reviewer feedback.
>
> >The experiment (compress to extreme) shows in Figure 2 is confounded.
>
> In the “extreme compression” experiment, we agree that the compression setting in Figure 2 simultaneously changes token count, spatial frequency bandwidth, aggregation, and positional information. Even when positional and fine-grained spatial cues are heavily degraded together with other factors, the model’s performance drops only slightly for many datasets. Since positional information is only one component among several being degraded, the fact that their joint removal causes minimal harm suggests that position alone is unlikely to be a dominant signal. The problem could be both model and the datasets, and it was the motivation for us to propose the 2DS dataset. As the contrast of Figure 2 we also put the results of the 2DS dataset in Figure 9. We can see a much more sharp drop on the 2DS with the same compression method. We will clarify this logic in the paper.
>
> >The link between Normalize and RoPE is indirect.
>
> For the link between Normalize and RoPE, we have the direct measure of layer-wise RoPE sensitivity in Sec. 6.4 Figure 8 and show that after Normalize a fixed positional perturbation induces substantially larger changes, indicating that RoPE becomes more consequential. We agree that +Multilayer likely improves performance partly by providing richer visual features, and we already discussed this in Sec. 7 and Appendix H. Our intent is to improve the spatial information perception capabilities of the model, and we present two approaches. Normalize is rebalancing norms to “unsuppress” RoPE, and +Multilayer is a complementary enhancement that improves spatial information awareness through additional visual features.
>
>
> >continuous control study
>
> For continuous control, we appreciate the suggestion for a continuous control study. Our current experiments already provide a direct diagnostic by measuring RoPE sensitivity across layers before and after normalization, which we believe is informative already.

---

> > ### Author Response · Authors · 2025-11-22
> >
> > >VQAv2/GQA are general
> >
> > We agree that VQAv2 and POPE are not dedicated spatial benchmarks, and in our paper they are used primarily as semantics-focused baselines. We respectfully disagree that GQA is not designed for left/right/front/back/orientation/reference. It includes a substantial subset of questions about left/right, in front of, behind, on top of, etc [1]. In addition for spatial reasoning we have CV-Bench 2D, which is specifically designed to test 2D spatial relations, and 2DS, which tests 2d spatial relationships directly. As shown in Appendix E.2, the largest gains from Normalize and Normalize+Multilayer appear on this spatially-focused group (GQA, CV-Bench 2D, 2DS), whereas VQAv2 and POPE see smaller but still positive improvements. This pattern is consistent with our claim that the interventions particularly benefit spatial reasoning.
> >
> > We appreciate the suggestions of What’sUp and GSR-Bench. GSR-Bench is, to our understanding, an extension of What’sUp and not yet publicly released, so our focus is on What’sUp.
> > Following the reviewer’s suggestion, we evaluate our models on the What’sUp benchmark. Below we report average accuracies over three seeds for each split:
> > | Split          | LLaVA-1.5 | +Normalize | +Normalize+Multilayer |
> > |----------------|-----------|-----------|------------------------|
> > | What’sUp Controlled A   | 58.66     | 76.94     | 72.82                 |
> > | What’sUp Controlled B   | 69.20     | 73.69     | 80.31                 |
> > | COCO Spatial-ONE       | 94.42     | 92.81     | 92.66                 |
> > | COCO Spatial-TWO       | 88.79     | 89.55     | 80.91                 |
> > | GQA Spatial-ONE        | 96.44     | 97.01     | 96.81                 |
> > | GQA Spatial-TWO        | 89.46     | 85.68     | 86.02                 |
> >
> > The What’sUp Controlled A and Controlled B splits are explicitly designed to probe spatial relations under controlled appearance. Here our interventions provide large and consistent improvements. Particularly we see +Normalize has +18.28% compared to LLaVA 1.5 in controlled A and +Normalize+Multilayer has  +11.11% compared to LLaVA 1.5 in controlled B. We believe the performance is consistent with our mechanism, that our interventions restore spatial signals in clean and isolated spatial reasoning datasets. We do notice COCO/GQA splits have mixed results. Both datasets achieve near-ceiling accuracies (both are binary choices).
> >
> > >The symbols in the figures
> >
> > For Figure readability, thank you for pointing out the readability issue. We will update the Figure 7 to improve the readability.
> >
> > [1] Hudson, Drew A., and Christopher D. Manning. "Gqa: A new dataset for real-world visual reasoning and compositional question answering." Proceedings of the IEEE/CVF conference on computer vision and pattern recognition. 2019.

---

### Official Review · Reviewer_119d · 2025-11-10

**Soundness:** 2
**Presentation:** 2
**Contribution:** 2
**Rating:** 4
**Confidence:** 3

**Summary:**

This paper investigates why Vision-Language Models (VLMs) like LLaVA can underutilize spatial information despite having positional encodings. The authors identify embedding norm imbalances between vision and text tokens as a key factor suppressing RoPE effectiveness. They introduce three interpretability tools and a synthetic dataset to measure spatial awareness, then propose two interventions: normalizing vision embeddings and incorporating intermediate vision encoder layers.

**Strengths:**

1. The paper is well-written and presents the idea clearly.

2. The paper attempts to address an important limitation in current VLMs regarding spatial reasoning capabilities.

3. The embedding norm analysis provides clear empirical evidence of the magnitude disparity between vision and text tokens.

4. The controlled 2DS dataset isolates spatial reasoning from semantic shortcuts, though with limitations.

**Weaknesses:**

1. Insufficient reproducibility details: Despite promising code release "upon publication," the paper lacks crucial implementation details. For example, the RMS normalization target values appear without justification; the H2 intervention also appears with very little detail and justifications for the choice of layers, etc.

2. Questionable dataset choices for analysis: The paper uses COCO validation set for most analyses (Figures 3, 4, 6, 7), but COCO is primarily object-centric rather than spatially-focused. This choice undermines claims about spatial mechanisms. Why not use spatially-rich datasets like various Spatial-VQA datasets [1, 2] or even their own 2DS throughout?

3. Limited evaluation of 2DS dataset: While 2DS aims to isolate spatial reasoning, it's overly simplistic (colored shapes in grids) and may not reflect real-world spatial complexity. Also, the paper doesn't validate whether improvements on 2DS transfer to natural images or whether the dataset truly eliminates semantic shortcuts as claimed. Again, why not using well-established Spatial-VQA benchmarks?

4. Attention head weights do not mean much in causality. As an interpretability work, I would assume it is trying to run intervention studies on salient attention heads or providing more interesting findings in the information flow, rather than merely saliency maps.

5. Statistical rigor lacking: Some results lack error bars or significance tests. Table 2's PSI differences could be noise.

[1] Chen B, Xu Z, Kirmani S, et al. Spatialvlm: Endowing vision-language models with spatial reasoning capabilities[C]//Proceedings of the IEEE/CVF Conference on Computer Vision and Pattern Recognition. 2024: 14455-14465.

[2] Zhang W, Zhou Z, Zheng Z, et al. Open3dvqa: A benchmark for comprehensive spatial reasoning with multimodal large language model in open space[J]. arXiv preprint arXiv:2503.11094, 2025.

**Questions:**

See Weaknesses. Also,

1. Why use COCO for spatial mechanism analysis when it's not designed for spatial reasoning? Have you validated your findings on truly spatial datasets?

2. What is the reason for directly correlating modality imbalance with attention heads? Honestly, I don’t find this transition very intuitive in the writing: “Functional behavior in Transformers is often head-specific, for example there are induction, copy, memory, summary, truthfulness attention heads in LLM. We therefore work at head granularity to give a closer look at the vision and text modality balance.”

---

> ### Author Response · Authors · 2025-11-22
>
> Thank you for the detailed and constructive review. We address your concerns point by point below.
> >Insufficient reproducibility details
>
> For reproducibility, we apologize that some implementation details were too condensed. Our goal is to match the scale of vision token norms to that of text tokens as seen by the LLM. Concretely, we first measure the L2 norms of text embeddings from the LLM’s embedding layer from random texts. The resulting distribution has mean ≈ 0.83 and max ≈ 1.22 (also reported in the main paper). For the H2 setting, we select CLIP encoder layers 12/16/20/24 based on prior work, which we cited right after the selected layers (Jiang et al., 2024) on line 319.
>
> >Questionable dataset choices for analysis
>
> For COCO dataset choice, our COCO analyses (Figures 3, 4, 6, 7) target internal representation statistics (token norms, logits, attention patterns), not task performance, and therefore do not rely on spatial labels. COCO provides a standard, reputable, and large corpus of image–caption pairs. The quantities we analyze (vision/text L2 norms, norm ratios, RoPE-affected logits) are functions of the model and inputs only. They do not depend on whether the underlying task is “spatial” or “semantic.” Using a spatial QA dataset would yield the same kind of norm distributions, but with fewer examples. To directly address your concern, we additionally computed token norm statistics on our 2DS spatial dataset and observed the same large modality gap corresponding to Figure 3. For example, on 2DS we obtain:
>
> -Vision token norms: mean 47.71, std 52.63
>
>
> -Text token norms: mean 0.74, std 0.22
>
>
> -Mean ratio (vision/text): 64.21
>
> It is in the similar range as the COCO. It shows that the choice of the dataset does not affect the internal model statistics.
>
> >Limited evaluation
>
> For the limitations of 2DS, we fully agree that 2DS is intentionally simple (colored shapes on grids). Its purpose is not to be a realistic benchmark but to serve as a controlled testbed. We want to ensure spatial relationships are the only way to solve the task and the ability to isolate the effect of representations (such as color and shape). We have already evaluated on 2 additional natural-image benchmarks including GQA and CV-Bench 2D, which both contain spatial questions. Our normalization and multilayer variants yield consistent improvements on these datasets (see Table 3 and Appendix E.2), indicating that the gains are not confined to the toy setup. On CV-Bench 2D in Appendix E.2 Table 7, improvements concentrate on the Relation category (spatial questions). This aligns with 2DS, where the largest gains arise on relative spatial queries. We appreciate the suggestion also to test established Spatial-VQA benchmarks such as SpatialVLM/Open3DVQA, but as the limitation discussed in Section 7, the 3D benchmark is not our focus, as position embedding in LLM could only differentiate 2D relative positions not 3D.

---

> > ### Author Response · Authors · 2025-11-22
> >
> > >Attention head weights do not mean much in causality.
> >
> > We agree that attention weights alone are not causal explanations. The main causal story in the paper is about embedding norm suppression of RoPE. The head-level attention visualizations (Figure 7) are presented as supplementary evidence that we find to be behaviorally interesting. We also discussed explicitly in line 408 that the larger vision share does not mean better performance, which means there might not be causality of large vision share and better vision.
> > What we present is a correlation to show whether the model focuses more on text or image, and how this correlation develops through layers.
> >
> > For correlation of modality imbalance with head-level patterns, transformer interpretability literature has repeatedly shown that functional roles are often head-specific (induction heads, copy heads, etc.) [1, 2]. Following that line, we analyze modality balance per head (fraction of attention mass on vision vs. text tokens) to see which heads are effectively focusing on vision after our interventions. This is meant as a descriptive behavior difference  instead of a causal claim about individual heads. We believe it is an interesting direction for future research and we do not intend to present it as a causal relationship.
> >
> > >Some results lack error bars or significance tests. Table 2's PSI differences could be noise.
> >
> > Since PSI is derived from accuracy before vs. after permutation, the natural place to test significance is on the underlying accuracy changes for each model/dataset pair. We therefore conduct paired significance tests comparing pre- and post-permutation performance. For POPE, GQA, CV-Bench 2D, and 2DS, where we have per-example predictions, we use the McNemar test. For VQAv2, evaluated via an external server (only aggregate accuracies available), we use a two-proportion Z-test.
> >
> > | Dataset      | LLaVA-1.5    | + Normalize  | + Normalize + Multilayer |
> > |-------------|-------------:|-------------:|-------------------------:|
> > | VQAv2       | p < 0.0001   | p < 0.0001   | p < 0.0001               |
> > | POPE        | 0.61         | 0.40         | 0.23                     |
> > | GQA         | p < 0.0001   | p < 0.0001   | p < 0.0001               |
> > | CV-Bench 2D | 0.66         | p < 0.0001   | p < 0.0001               |
> > | 2DS         | p < 0.0001   | p < 0.0001   | p < 0.0001               |
> >
> > As expected, POPE shows no significant difference (it does not directly probe spatial reasoning), and baseline LLaVA-1.5 on CV-Bench 2D is also not significant before vs. after permutation. In contrast, for spatially focused datasets (GQA, CV-Bench 2D with our interventions, and 2DS) and for VQAv2, the differences are highly significant across models, indicating that the PSI changes we report are not attributable to noise.
> >
> > [1] Olsson, Catherine, et al. "In-context learning and induction heads." arXiv preprint arXiv:2209.11895 (2022).
> >
> > [2] McDougall, Callum, et al. "Copy suppression: Comprehensively understanding an attention head." arXiv preprint arXiv:2310.04625 (2023).

---

> > > ### Comment · Reviewer_119d · 2025-11-28
> > >
> > > Thank you for the detailed response. I appreciate the additional implementation details and statistical significance tests, which strengthen the validity of the work!
> > >
> > > However, I still remain concerned about the framing of the attention head analysis. I agree with the statement that attention weights are for showing correlation, yet the current manuscript frames this as understanding "how the VLMs integrate the vision and text modality" (line 378). This is a mechanistic claim, not a correlational observation. Furthermore, while you cite literature on several head-specific functions (mostly relevant to in-context learning), there is a need for theoretical or empirical supports that **spatial reasoning would similarly manifest as head-specific patterns**. The leap from "some functions are head-specific" to "let's analyze spatial awareness at head granularity" lacks careful justification.
> > >
> > > If the contribution is mainly about embedding norm suppression of RoPE, then the paper could focus on that mechanism with appropriate causal evidence, rather than presenting extensive attention visualizations with grand interpretive claims.

---

> > > > ### Author Response · Authors · 2025-11-30
> > > >
> > > > Thank you again for the thoughtful and helpful response, and we really appreciate the feedback. We agree that the central contribution of the paper is the embedding-norm suppression mechanism. The attention head visualizations were included because we found the behavior differences under our interventions interesting and potentially valuable for future work, and we were excited to share them. However, we agree with your assessment and will revise the framing accordingly.
> > > >
> > > > In the revision, we will (1) remove the mechanistic language you pointed out and clearly describe the head-level analysis as **correlational and exploratory**, and (2) re-center the Section 6.3 to on our main contribution: the **embedding norm suppression of RoPE** and the associated norm-balancing interventions, supported by RoPE-sensitivity, PSI, and spatial benchmark results.

---

### Meta-Review · Area_Chair_sndT · 2025-12-22

**Summary:**

The submission aims to discover spatial awareness in vision-language models.  However, reviewers are concerned about the validity of the analysis, their applicability to models beyond LLaVa and RoPE, as well as various presentation and reproducibility issues.

**Reviewer Concerns:**

The concern about the applicability to models beyond LLaVa is addressed.  However, most other concerns remain unaddressed.

**Reviewer Scores:**

The reviewers would have maintained their scores of 4, 2, 2, 4.

---

### Decision · Program_Chairs · 2026-01-26

Reject